# Effectiveness of Community Occupational Therapy Intervention in, with and from the Community in People with Disabilities in Azrou (Morocco)

**DOI:** 10.3390/ijerph18115602

**Published:** 2021-05-24

**Authors:** Jerónimo J. González-Bernal, Leire Eiguren-Munitis, Josefa González-Santos, Mirian Santamaría-Peláez, Raúl Soto-Cámara, Paula Rodríguez-Fernández

**Affiliations:** Department of Health Sciences, University of Burgos, 09001 Burgos, Spain; jejavier@ubu.es (J.J.G.-B.); mspelaez@ubu.es (M.S.-P.); rscamara@ubu.es (R.S.-C.); prfernandez@ubu.es (P.R.-F.)

**Keywords:** community Occupational Therapy, significant activity, inclusion, Morocco

## Abstract

In Morocco, the social and environmental context influences the volition and development of meaningful activities, creating physical, personal and social barriers to the occupational performance of people with disabilities. This study develops a community Occupational Therapy program in order to verify its effectiveness in the volition, quality of life and perceived self-stigma of people with disabilities in the Moroccan city of Azrou, and to reduce the stigma of the community towards people with disabilities in the city. Data were collected from people with disabilities who participated in the program (N = 52), using the Volitional Questionnaire (VQ), The World Health Organization Quality of Life scale (WHOQOL-BREF), the Stigma Awareness Questionnaire (SCQ) and an ad hoc interview. In addition, community stigma was assessed by the Attribution Questionnaire (AQ-27) in citizens without disabilities (N = 42). Results confirmed that this intervention favors the inclusion of people with disabilities in their closest environment, improving volition and quality of life and reducing self-stigma. Furthermore, the community’s stigma towards people with disabilities was also significantly reduced after the intervention.

## 1. Introduction

Disability is defined by the WHO [1] as a complex phenomenon that reflects a close relationship between the characteristics of the person and the environment, being determining of the sociocultural properties. According to the latest WHO results [1], 15% of the world population has some type of disability, and the Moroccan National Survey [1,2] confirms that 5.12% of its population has a diagnosis of disability, or as far as this is concerned, more than one and a half million of the total population. Motor (50.2%) and mental (25.1%) diseases are the most diagnosed, compared to the communicative (10.5%) and metabolic (8.5%) ones [2,3].

According to Morocco’s State Survey [2], the situation of people with disabilities is confusing. Although there are laws in the regulatory framework that underscore and ensure the rights and inclusion of people with disabilities in their community, the reality that many of them live in is very different [3]. Therefore, actions must be taken to ensure compliance with rights and thus achieve personal and community well-being and social participation.

However, despite the fact that there are measures that favor the inclusion of people with disabilities in the public sector, the state report shows that the unemployment rate among this group is five times higher than in the rest of the population [2,3]. In the educational field, 85.7% of people with disabilities has not received any kind of legal education and, due to the inadequacy of specialized health infrastructures, access to social and health services is impossible for 60.8% of the population [2,3]. As a result, the leading causes of disabilities as well as the treatments available to people with disabilities are unknown [2,3]. According to the same report [2,3], more than 43% of respondents attribute the cause of their disability to evil, magical or divine origins.

With regard to the normative concerning disability, over the past 50 years, parliament has made many efforts to establish laws to ensure access to rights and social participation, highlighting Law No. 97/13 of 1993 [4,5,6,7,8], which guarantees social protection and its subsequent adaptation in 2014; the accessibility law; and a labor code that ensures workers’ rights [4,5,6]. In addition, the signing of the United Nations International Convention on the Rights of People with Disabilities [4,5,6,9,10] includes the rights and obligations of states to safeguard equal opportunities. Despite this, the lack of sensitivity towards this group continues to be a reality. Most of the affected people and their families do not receive any kind of social or economic assistance from governments and institutions, and it is the relatives themselves or close people who place limits both on the development of their autonomy [11], and on the performance of activities and participation in society [12,13]. This results in a situation of occupational deprivation, which is understood as a prolonged restriction of participation in significant activities due to due to outside circumstances [11,14]. 

In contexts where health is conceived as equity and social participation [15], the community, understood as a social network that contributes to the determination of integration and social support, gives special relevance to time to design and carry out interventions to promote the social inclusion of people with disabilities in Morocco [16]. At the same time, from Occupational Therapy (OT), a discipline that promotes the occupational health of the entire population, it is necessary to develop a decolonizing (without imposing the criteria of a dominant culture) and decolonizing (with its own theory and practice, not so linked to the biomedical field) theory and practice that moves away from the hegemonic academy and develops culturally-sensitive and effective interventions [17,18,19].

Therefore, the community approach to OT proposes three perspectives that, in addition to favoring the participation of all members of society in the therapeutic process, take into consideration the perspective of health as equity and social participation [20]. OT in the community arises with the aim of improving access and occupational participation of all members of the community. This approach is influenced by the Customer Centered Practice Model [21] that was born with the aim of responding to the disadvantages of the biomedical model. In this way, OT in the community requires a structural change to address the needs of clients and does not focus attention on the objectives of professionals [22]. This approach involves interventions for community development, giving particular importance to the context, barriers and policies of the community [23]. Community OT is influenced by the Community-Based Rehabilitation Model [24], which aims to promote the better distribution of community resources, it being the therapist’s role to establish partnerships and negotiations with the territory’s bodies, as well as to promote community development of people with disabilities [25]. Finally, community therapy is related to the Independent Living Model [26], which offers a different approach to persons with disabilities by promoting the right to equal conditions and encouraging decision-making by people who are disadvantaged [27].

As Kronenberg and Simó defend in their project “Occupational Therapy Without Borders” [13], the community approach to address the occupational problems of the general population responds to aspects that are outside the social welfare system. In this process, the therapist must acquire different roles, such as advocate, case manager and consultant, entrepreneur, supervisor, community program coordinator, researcher and teacher [28,29]. Thus, taking into account the situation of people with disabilities in northern Morocco, the approach of interventions in, with and from the community could promote awareness and reduce exclusion towards disability.

This investigation had two main objectives. On the one hand, the aim of this research was to verify the effectiveness of a community based OT intervention in the volition, quality of life and perceived self-stigma of people with disabilities in the Moroccan city of Azrou. On the other hand, another main objective was to reduce the stigma of the community towards people with disabilities in Azrou. Therefore, the research question was whether a community based OT intervention was effective to improve the volition, quality of life, awareness of stigma and occupational performance of people with disabilities, as well as to reduce the stigma of society towards this group.

## 2. Materials and Methods

### 2.1. Study Design—Participants

Longitudinal study conducted through the focus of working on, with and from the community and based on the development of meaningful activities. The action research process lasted 5 months, and it was carried out with two groups; the first one was composed of people with disabilities that participated in an intervention program (N = 52), and the second one (N = 42) was formed by people without disabilities from the community where the intervention was developed. 

The criteria for participating in the group of people with disabilities in the research were to be a beneficiary or to be in contact with the “Happiness Without Borders (HWB) Association” and to reside in Azrou or in the continuous localities. Thus, people who were not linked to the Association or who participated and collaborated with other non-governmental entities were excluded from the study. For participating in the group of people from the community, the criteria were not having any disability and to reside in Azrou.

### 2.2. Procedure

The participants were recruited through convenience sampling, and the methodological design was carried out under the ethnographic approach which, according to Hammersley and Atkinson [30], is the most basic form of social research and requires direct contact with the social agents, recording experiences in a respectful way [31]. It is an approach characterized by the researcher´s participation in the daily life of the people involved in the study, which is essential to know the cultural meanings of the context.

Regarding the group of people with disabilities, participants were selected by the main researchers of the project and the president of HWB, considering those who regularly attended the project called “Rehabilitation and Inclusion”. After this previous work, people with disabilities susceptible to intervention were informed about the objectives and procedure of the investigation, as well as the possible risks and benefits that could derive from it, after which the corresponding informed consents were signed. 

Access to people in the community was through their participation in local events and their presence at strategic points of the city, such as the Souk or the Mosque. They were also informed of the objectives and procedure of the research. 

After the recruitment of the participants from both groups, the initial evaluation was carried out, prior to the implementation of the intervention in the group of people with disabilities for 16 weeks. People in the community group did not participate in the intervention, although it was developed in their own community, and they were able to see what was being done. One month after the intervention, all variables were re-evaluated for further analysis in both groups.

The Bioethics Committee of the University of Burgos approved the research (Reference IR 7/2018), which was carried out according the ethics principles of the Helsinki Declaration [32], the Regulation 2016/679 of the European Parliament and of the Council of 27 April 2016 [33], The Organic Law 3/2008 of December 5, protection of personal data and guarantee of digital rights [34], the ethical code of OT [35,36] and the regulations established by the European Commission by the European Group on Ethics in Science and New Technologies [37].

### 2.3. Intervention Process

Prior to the implementation of the intervention, volition, quality of life, awareness of stigma and other relevant variables in the ad hoc questionnaire were evaluated in the group participants with disabilities. Due to their characteristics and the conception of health as something collective, the study variables were evaluated through informants, the caregivers, usually mothers and grandmothers, who provided the information in most cases. Regarding the group of participants without disabilities, no tool adapted to the rural population of Morocco was found, so a meeting was held with 68 people from Azrou and the continuous villages in which the different ways of evaluating the stigma of society towards disability were discussed. It was decided that the AQ-27 questionnaire, despite being aimed at the population with mental illness, was the most appropriate tool after a few simple adaptations. Once adapted, the questionnaire was piloted in a sample of 50 subjects to check its validity and ensure its adaptation to the sociocultural context, to later be completed by the 42 people who made up the study group.

Initial evaluations were performed in both groups, and then intervention sessions were carried out in the group of people with disabilities, from a community approach and based on the realization of significant activities focused mainly on games and leisure. The intervention lasted 16 weeks, during which one-hour sessions were held, 7 days a week. 

Individual and group sessions were developed, always with the help of a translator, to facilitate communication between the therapist and the participants. The sessions were elaborated, taking into account the responses of the ad hoc questionnaire, in order to consider the customs and socio-cultural characteristics of the place and the needs and interests of the participants. The intervention activities were agreed upon with the interested person and their closest circle and were planned and developed based on their significant occupations, collected through the previously administered ad hoc questionnaire. The activities were graduated according to the person´s needs and were initially developed in a small and structured environment to be later generalized to real contexts, such as participating in family businesses, enjoying a space and leisure activities, or collaborating in activities religious. During the performance of these occupations, in addition to promoting inclusion in the community, individual performance skills such as motor skills, emotional skills, cognitive skills, communication skills and social skills were promoted. By holding the sessions in the community environment, the rest of the population also participated indirectly in the different sessions, which was intended to reduce the stigma of the community and promote inclusive behaviors and thoughts towards this group.

One month after the end of the intervention, a second evaluation was carried out with both groups in order to conduct the subsequent analysis.

### 2.4. Instruments

The data were collected using several standardized scales and an ad hoc interview through participating observation method. The scales made it possible to collect specific data with good statistical value, while the ad hoc interview provided a more global view of the phenomenon studied. Relevant sociodemographic variables such as gender or age were also collected for further analysis. The entire data collection process was carried out in habitual and natural contexts with the aim of observing, listening and maintaining an informal conversation with participants during the development of daily activities and tasks, creating and annotating field notes after observation through reflections with participants. 

The evaluation tools used for the group of people with disabilities were the following:The Volitive Questionnaire (VQ), which aims to evaluate the volition according to the Human Occupation Model, referring to one of the subsystems of the person [38]. It consists of 14 items that provide individualized and specific information about the person’s motivational characteristics, activities and other circumstances that increase the person’s volition, evaluated by a scoring system consisting of P (passive) = 1, D (hesitant) = 2, I (involved) = 3 and E (spontaneous) = 4 [39,40,41]. Thus, it rates from 4 to 56, and higher score means higher volition.The World Health Organization Quality of Life scale (WHOQOL-BREF) [42]. It allows one to know the general profile of Quality of Life, through 26 items divided into four dimensions (physical, psychological, social relations and environment) through a Likert type scale of 5 points ranging between 26 and 130, where higher scores mean a better quality of life.The Stigma Consciousness Questionnaire (SCQ) [43]: to know the self-stigma of people with disabilities, through 10 items with a Likert scale with 5 response options, 1 being not agree and 5 totally agree. Three items are inverse, and so they were recodified for data evaluation, so it scores from 5 to 50, where the higher scores mean the lower stigma consciousness or auto-stigma.Ad hoc interview, divided into two blocks, in which the former collects information on cognitive level and occupational behavior with responses inadequate = 0 and appropriate = 1; emotional state, communication and interaction skills and social functioning with a 1–5 Likert scale; and the activities of daily life (ADL) with values dependent = 0 and independent = 1; in all cases, higher scores are assimilated to better circumstances. The second block consists of four open questions translated and adapted to the cultural and linguistic needs of the country on the significant activities of the person, which was used to plan the significant activities used in the intervention.

All participants of the group of people without disabilities were evaluated by the Attribution Questionnaire (AQ-27) [44], adapted to the context. The original AQ-27 questionnaire assesses the stigmatizing attitude and beliefs towards people diagnosed with mental illness; however, for this research, questions were tailored to the context and collective to be analyzed, emphasizing stereotypes, beliefs and behaviors towards people with disabilities. It consists of 27 items distributed in 9 factors (blame, anger, pity, help, dangerousness, fear, avoidance, segregation and coercion) scoring between 1 and 9, where avoidance items have an inverse score and higher scores mean a higher stigmatizing attitude and beliefs towards people with disabilities [45]. Each dimension consists of 3 items and scores between 3 and 27. Scores greater than 20 are considered high, between 20 and 10 average, and less than 10 low [44]; the total score of the scale can also be considered [46].

### 2.5. Statistical Analysis

A normality analysis was performed using the Kolmogorov–Smirnoff test (N = 52) and the Sapphire–Wilk test (N = 42), which determined that the sample did not fit a normal distribution. Descriptive analysis was performed in terms of mean and standard deviation (SD) in continuous variables and percentages and frequencies in categorical variables. For the inferential analysis, Spearman correlations were performed with the quantitative variables and the Wilcoxon rank-sum test to compare the scores obtained in the pretest and in the post-test in both study groups. 

For statistical analysis, it was conducted using SPSS Version 25.0 (IBM Corp, Armonk, NY, USA) software, with a set risk of 0.05 as the limit of statistical significance.

## 3. Results

### 3.1. People with Disabilities Participating in the Intervention Program (N = 52)

Out of the total number of people with disabilities participating in the intervention program (N = 52), there were 29 men (55.8%) and 23 women (44.2%), with an average age of 12.08 (SD ± 6.52) years old. Most people were single (N = 42; 80.8%), and regarding the residence area, 69.2% lived in a rural area (N = 36) and 30.8% in an urban area (N = 16). Most had preschool studies (N = 24; 46.2%) or primary (N = 21; 40.4%), and only four people had secondary studies (7.7%) and three subjects had higher education (5.8%).

In terms of disability, most had a physical disability (N = 23; 44.2%), 14 people had mental disabilities (26.9%), 3 people emotional (5.8%), 9 mixed disability (17.3%) and 3 subjects of unknown type (5.8%) (Figure 1). Most participants had a primary caregiver (N = 41; 78.8%), whether a family member or a professional. Additionally, most people moved independently (N = 41; 78.8%) and only 11 participants needed mobility aid (21.2%). In addition to the main diagnosis of their disability, there were participants with addictions (N = 14; 26.9%), family conflicts (N = 11; 21.2%), communication difficulties (N = 8; 15.4%) and close fights (N = 3; 5.8%). Sixteen people did not present added difficulties (30.8%).

In inferential analysis, the age variable was negatively correlated with volition in leisure activities (r (52) = −0.326, *p* = 0.018) and at home activities (r (52) = −0.303, *p* = 0.029), as well as with the quality of life prior to the intervention (r (52) = −0.476, *p* < 0.001), so that the older participants had less volition and a lower level of quality of life. 

Statistically significant differences were found between the scores before and after the intervention in all the independent variables studied, which are reflected in Table 1.

### 3.2. People from the Community Who Completed the AQ-27 Questionnaire (N = 42)

Of the total number of people in the community (N = 42) who completed the AQ-27 questionnaire, 50% were men and 50% women, aged between 8 and 64 years (M = 31.31; SD ± 14, 75). Twenty-five people lived in a rural area (59.5%) and 17 in an urban area (40.5%). 

After inferential analysis, age did not correlate with stigmatizing attitude towards people with disabilities. Regarding the effectiveness of the intervention, significant differences were identified between the scores before and after the intervention, both in the total score of the AQ-27 scale and in each of its scores’ dimensions (guilt, anger, pity, help, dangerousness, fear, avoidance, segregation and coercion) (Table 2).

## 4. Discussion

This study aims to clarify whether intervention in, from and with the community based on significant activities is effective for improving the volition, quality of life and stigma consciousness in participants with disabilities, as well as to verify if this intervention, in turn, is able to diminish society’s stigmatizing attitude towards these people.

A significant increase in the volition of people with disabilities was found in the three areas of activity evaluated, as well as in quality of life. For its part, the stigma consciousness of people with disabilities themselves was lower after the intervention. Likewise, the variables analyzed using the ad hoc questionnaire also provided indicative results of improvement in cognitive level, emotional status, occupational behavior, ADL, communication and interaction skills and social functioning. Regarding the stigmatizing attitude of society towards people with disabilities, a significant reduction was found after the intervention. The OT theoretical basis consider that interventions based on the performance of specific activities or occupations in certain environments promote the participation of people in these settings and improve performance in these activities [47,48]. In this research, following the axiom that the impact on participation in the ADL has a great impact on their personal, functional, social and economic life [49,50,51], it was shown that this type of intervention can improve the performance of the participants with disabilities. Furthermore, the stigmatizing attitude of society can be reduced, thereby creating opportunities for community participation for these people.

Strengthening a model of socio-community inclusion for people with disabilities is a vital goal in a way that supports community inclusion, and people with disabilities can develop roles within their membership community; all this is to normalize people with disabilities for their socio-community inclusion [52]. Thus, OT intervention requires the inclusion of the community as an area of development of the person and as a context in which their occupational performance is framed and determined [53]. This study provides an intervention that is effective from this conceptualization, as the results support that such approaches can be beneficial to improve that occupational performance.

Community OT has been shown to be useful in improving occupational performance in children and youth in a study with 167 participants under the age of 18, so that children receiving home and community treatment gained changes in their performance skills [54]. These findings are consistent with those obtained in this study, since improvements were demonstrated in the variables evaluated in the ad hoc questionnaire, in addition to volition. A systematic review about the effectiveness of community OT interventions, which included 12 articles published between 2007 and 2020 [55], concluded that it constitutes a consolidated line of research, but with limited objectives and research areas, so that the studies were predominantly qualitative and descriptive. In addition, the studies analyzed showed an average/low level of evidence, among which the ones that intended to reduce the risk of fall and to improve performance in the ADL were those that showed the best results [55]. Along these lines, in this study the treatment seemed to be effective in improving all the variables studied.

Regarding volition, previous investigations associated this variable with greater participation in certain activities. An Australian study, involving 244 adults older than 70 years, demonstrated a significant association of volition with increased participation in physical activities in older adults living in the community [56]. Another study with adolescents with cerebral palsy demonstrated that occupational therapists can increase the motivation and will of this group through self-efficacy in interaction with environmental characteristics to promote personal causality [57]. Graff et al. [58] found in their study of 135 couples of people with dementia and their caregivers that community OT improves the mood, quality of life and health of both people with dementia and their caregivers, improvement that was significant in the follow-up after 12 weeks. Although this study does not consider the effects on caregivers, it does show similar results in terms of improving people´s quality of life. Carrasco, Martín and Molero [43] found a negative relationship between stigma consciousness and quality of life, with differences depending on sex, disability type and reason for disability, in a study that included 201 people with sensory, physical and other disabilities. Another study with children with disabilities and their families in Zambia [59] concluded that in low and middle-income countries, especially in low-income communities, the stigma of disability reduces their opportunities for participation, and they advocated for community-based interventions as a viable and acceptable approach to interacting with the community and families of children with disabilities. Although it was a pilot study, families and individuals in the community reported less perceived rejection from family and peers, but the agreement that children with disabilities should be treated in the same way as other children was minor [59]. These findings are consistent with this research, as there was less awareness of stigma on the part of participating people with disabilities, and a less stigmatizing attitude on the part of the community was observed after the intervention.

The provision of therapeutic services based on western practice in other types of communities causes challenges and conflicts because the differences between cultural beliefs, values and customs are significant and can be an important barrier to the development of interventions; therefore, one study concludes that it is necessary to develop an appropriate epistemology for non-western cultures that provides guidelines for successfully implementing such services tailored to the people of those communities [60]. More research about what kinds of strategies are successful in these non-western contexts through studies developed with and adapted to local populations are needed, such as this research. 

This study has both strengths and weaknesses. The implementation of a community intervention in a population with limited resources, and the good results in terms of the participation of both persons with disabilities and their families and the rest of the community, with the aim of improving social participation and reducing the stigma towards disability, is the greatest strength. Furthermore, the fact that there was a significant improvement in all the variables studied is encouraging and facilitates the approach of future similar lines of research in this or other communities. On the other hand, the design has methodological limitations that should be pointed out as weaknesses. The sample was reduced in people with disabilities and in the people who completed the AQ-27 questionnaire, using a non-probability sampling. In addition, there was no control group, so it could not be guaranteed that the changes that occurred were due to the intervention. Although standardized and validated tests were used to evaluate some aspects, in others it was not possible, hence the elaboration of an ad hoc questionnaire, which, despite not having the psychometric properties offered by the validated scales, provides relevant information and highlights the need to create appropriate assessment tools for non-western communities.

The present research provides evidence about the importance of community interventions in populations in which people with disabilities are subject to great stigma from society and from themselves. The community perspective allows professionals to address issues involved with people´s reality, generating a space for communication, dialogue and participation, and which also favors social cohesion. It is a continuous, dynamic and dialectical process that improves the living conditions of a certain community and permits the formation of non-hierarchical relationships, maintaining horizontality in power and knowledge. Future research with a larger and more representative sample is recommended, allowing the intervention to be extrapolated to other locations.

## 5. Conclusions

Most of the existing studies have been carried out from a western approach and in populations that are not closely related to the one investigated in the present study. It is necessary to carry out research in this type of population, so this study constitutes an interesting contribution by investigating aspects such as volition in the performance of different activities, quality of life and awareness of the stigma of people with disabilities. Additionally, it considers the stigmatizing attitudes of society towards these people. 

The intervention was positive in that it improved all the variables studied using standardized instruments, and, in relation to variables of the ad hoc questionnaire, the intervention was also effective in improving cognitive level, emotional state, occupational behavior, ADL, communication and interaction skills and social functioning. This study provides evidence that justifies community OT intervention and proposes more interventions and research in this line, which appear to be effective. In addition, it would be very convenient to replicate the results and establish studies that include a control group.

## Figures and Tables

**Figure 1 ijerph-18-05602-f001:**
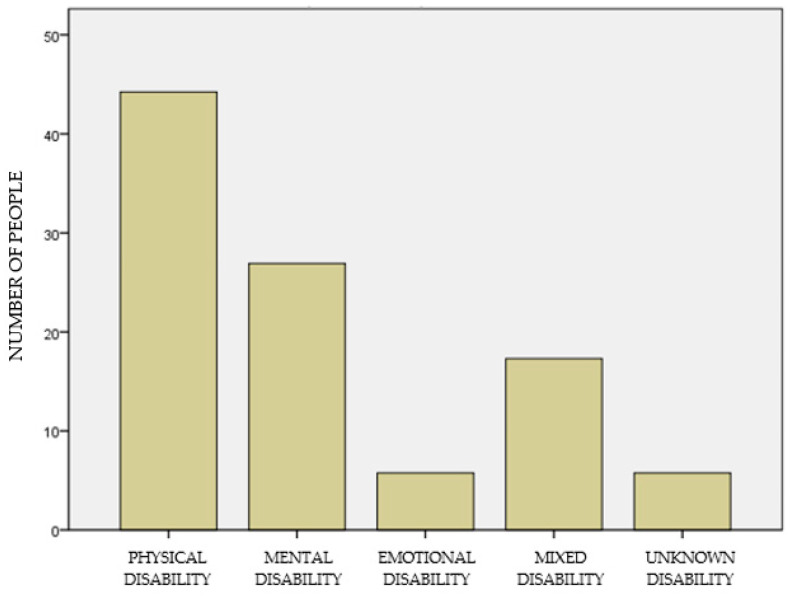
Type of disability.

**Table 1 ijerph-18-05602-t001:** Pre- and post-test comparisons in the group of people with disabilities.

Evaluation Tools		Mean			
		Pre-Test	Post-Test	Z	*p*-Value
VQ	Volition center	16.50	36.00	−6.282 ^a^	<0.001
	Volition leisure	14.00	27.00	−6.280 ^a^	<0.001
	Volition home	14.00	32.00	−6.278 ^a^	<0.001
WHOQOL-BREF	Quality of life	38.00	74.00	−6.290 ^a^	<0.001
SCQ	Stigma consciousness	34.50	28.00	−5.967 ^b^	<0.001
Ad hoc questionnaire	Cognitive level	3.00	6.00	−6.306 ^a^	<0.001
	Emotional status	8.00	18.00	−6.283^a^	<0.001
	Occupational behavior	3.00	7.00	−6.319 ^a^	<0.001
	ADL	4.00	9.00	−6.132 ^a^	<0.001
	Communication and interaction skills	10.00	20.00	−6.281 ^a^	<0.001
	Social functioning	7.00	14.00	−6.287 ^a^	<0.001

VQ: Volitive Questionnaire; WHOQOL-BREF: World Health Organization Quality of Life scale; SCQ: Stigma Consciousness Questionnaire; ADL: activities of daily life. ^a^ Based on negative ranges. ^b^ Based on positive ranges.

**Table 2 ijerph-18-05602-t002:** Pre- and post-test comparisons in the group of people without disabilities.

Evaluation Tool		Mean			
		Pre-Test	Post-Test	Z	*p*-Value
AQ-27	Blame	13.00	16.00	−4.469 ^a^	<0.001
	Anger	21.00	8.00	−5.655 ^b^	<0.001
	Pity	15.00	16.00	−3.765 ^a^	<0.001
	Help	10.00	20.00	−5.652 ^a^	<0.001
	Dangerousness	21.00	8.00	−5.655 ^b^	<0.001
	Fear	19.50	6.00	−5.669 ^b^	<0.001
	Avoidance	20.00	11.00	−5.627 ^b^	<0.001
	Segregation	24.00	9.00	−5.695 ^b^	<0.001
	Coercion	21.50	10.00	−5.653 ^b^	<0.001
	Total stigmatizing attitude	167.50	103.00	−5.646 ^b^	<0.001

AQ-27: Attribution Questionnaire. ^a^ Based on negative ranges. ^b^ Based on positive ranges.

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
