# Peer review of "Effectiveness of Community Occupational Therapy Intervention in, with and from the Community in People with Disabilities in Azrou (Morocco)"

_ijerph, 2021, doi:10.3390/ijerph18115602_

Round 1

Reviewer 1 Report

This is interesting work and I believe is of interest to the readers. I commend the researchers for their work. In the revision, just be mindful of the flow of the paper and to tighten how things are written. A more scientific approach is warranted as some areas for the methodology and results are lacking.  I do believe that this research has strong potential 

Here are my reviews: 

Abstract

Line 20 – 21, start sentence with Results confirmed that this intervention, if the authors want to save on words for the abstract

Introduction

Line 29, there is an extra space between the full stop and This..

Line 29 – 33 – this is a very long sentence, consider breaking it up and restructuring as it is confusing

Line 34 – 36 – Results from the national survey show that approximately 1.5 million (5.12%) of Moroccans …..

Line 34 – 37 – consider restructuring and adding to the first paragraph to make one paragraph.

Line 43 -44 – rephrase, (e.g As a result, the leading causes of disabilities as well as the treatments available to people with disabilities are unknown. According to the same report….

Line 47 – missing years after 50

Line 58 – 59 – explain in a sentence or 2 what is meant by occupational deprivation

Line 60 – 63 – another long sentence, consider breaking up

Line 63 – what is OT??? spell out before abbreviating

Materials and Methods

Participants

Line 92 – how long was the longitudinal study

Indicate inclusion and exclusion criteria

How were individuals selected to participate in the study?

How did the researchers come up with 52 people in one group and 42 in another?

Procedure

Line 104 – 114 – consider putting this in the section under participants

Why were the study variables evaluated by informants (mothers and grandmothers)?

Line 122-125 – consider putting this in the section under participants

A separate section should be formed called Intervention where the authors outline how they developed the intervention

Did the participants give informed consent? Were they compensated, if so how much?

Was this a pre and post test? Were participants given the surveys, then the intervention and then given the surveys again?

Was the intervention informed based on the methodology put forth by community based participatory intervention??

Ad-hoc semi-structured interview – I am not sure why in the qualitative portion of this, the interviewers will still use likert scales. The only semi-structured part of this is the open-ended questions

How did the researchers adapt the AQ-27 scale? Did they do a small pilot study to assess validity?

Line 177 – 179 – consider including this in a specific section for development of the intervention

How long after the implementation of the intervention did researchers administer surveys?

Did researchers have follow up surveys at least 3 months later to determine if the effects of the intervention prolonged?

Results

I am not sure why the researchers are talking about a correlation between age and stigmatizing attitude. Nothing of this sort was mentioned in the objectives of this study

Did the researchers have any hypotheses?

Are there any significant differences between the two group on any of the variables assessed?

The authors mentioned that an objective is to reduce the stigma of the community towards however the results and analysis did not show that this was assessed

Where are the results for the ad hoc semi-structured interview particularly for the open-ended questions

Discussion

Line 258 – what is ADL?? Spell it out

Line 271 – 283: I get why the researchers mentioned this study, however the purpose of their study was to test disability vs non disability and not to test the setting across different interventions. Consider finding a different study to show support of your results. Perhaps even stress how novel this study is and what in can do for the community and people with disabilities.

Line 285 – with Australian study – same applies, sample was kids, this study is older adults

Consider a section for practical implications

The authors should also think of including some of these studies in the introduction section.

Re-iterate why your study is important and what it means for the wider society in the discussion. Also consider putting future research suggestions in the section under limitations.

Reviewer 2 Report

I have read the article entitled “Effectiveness of community occupational therapy intervention 2 in, with and from the community in people with disabilities in 3 Azrou (Morocco)”. The overall novelty of this work is good, but the results are not well presented. I cannot, therefore, recommend this paper for publication in International Journal of Environmental Research and Public Health in the present form.

  • make a graph with the disability results
  • What occupational therapies are needed in this population and why?
  • how therapeutic services are adapted for non-western cultures, explain and discuss
  • write the advantages of occupational therapy intervention
  • add more information to the introduction

Reviewer 3 Report

This paper can be useful for other researchers in this area of using occupational therapy to disability groups somewhere in the world, especially in developing countries. 

My remarks are the following. Answering these remarks by improving the text in these fields will improve the paper to a interesting an useful paper . 

1

You write 5% have a disability according to a national reference. I think you should compare or refence also to a global source like the WHO disability numbers like on Disability and health (who.int) that shows a number of 15% in general. Please add a few sentences about the differences depending on the sources, so the reader will understand that the statistics differ depending on who is counting and on the definition of what is a disability.

2

I  appreciate you used research tools that are published by independent sources like the Likert scale, the questionnaires, etc.

But about the OT intervention it is not clear to me and I suppose also not for the readers what the OT intervention in your study means in terms of hours or days of investment and actual task of the persons involved. For example was it a course of 10 days each 6 hours meeting like having a class where an occupational therapist was lecturing about roles of different formal occupations from the local government? Or was is observing in the houses and interviewing the individuals to see what they needed and wanted to become an independent  citizen and how to reach that objective and to show what it would cost ? 

3

About the sample: you wrote it is a non-probabilistic convenience sample. This is a little bit double formulated (tautology or pleonasm) because a convenience sample means already it is a easy taken sample which is not representative for the population. In addition I would like to know if this sample was not too easy taken, for example not 5 families of 10 people or a group of people from the same street or church . So can you tell the reader there is no relation between the members of the sample? they are random found trough an advertisement in a local newspaper or how else did you recruit the sample? 

Round 2

Reviewer 1 Report

Thank you to the authors for addressing the comments. All my comments have been sufficiently addressed and I am pleased with how the authors addressed them.